# Effect of Increased Depressive Feelings during the COVID-19 Pandemic on the Association between Depressive Mood and Suicidal Behavior: Using the 17th (2021) Korea Youth Risk Behavior Web-Based Survey (KYRBS)

**DOI:** 10.3390/ijerph192214868

**Published:** 2022-11-11

**Authors:** Jinsu Yeo, Geunhyoung Park, Joohyun Shin, Kwanghyun Seo, Yeonsoon Ahn

**Affiliations:** 1Yonsei University Wonju College of Medicine, Wonju 26424, Korea; 2Department of Occupational and Environmental Medicine, Wonju Severance Christian Hospital, Yonsei University Wonju College of Medicine, Wonju 26426, Korea; 3Department of Preventive Medicine and Genomic Cohort Institute, Yonsei University Wonju College of Medicine, Wonju 26426, Korea

**Keywords:** depressive mood change, COVID-19, suicidal behavior

## Abstract

This study investigated the impact of increased depressive feelings during the COVID-19 pandemic on the suicidal behavior of Korean high school students using the 17th Korea Youth Risk Behavior Web-Based Survey (KYRBS). We classified the independent variables into four groups (“no depressive mood and no increase in depressive feelings [group A]”, “no depressive mood and increase in depressive feelings [group B]”, “depressive mood and no increase in depressive feelings [group C]”, and “depressive mood and increase in depressive feelings [group D]”). Compared to group A, group D showed an odds ratio of 18.30 in men and 14.87 in women for suicidal behavior after accounting for demographic and health behavioral characteristics. We found that depressive mood and a relatively short-term increase in depressive feelings had a synergistic effect, rather than an additive one, on suicidal behavior. Based on this result, we claim that an appropriate intervention method is necessary to prevent adolescents’ depressive mood from developing into suicidal behavior during the pandemic.

## 1. Introduction

The coronavirus disease (COVID-19) pandemic has caused many changes in our lives. Many countries, including Korea, have implemented lockdown and social distancing policies, leading to an increase in online activities and a decrease in private gatherings. Owing to these daily changes, the number of people reporting depression has doubled since the beginning of the pandemic [1]. This shows that the isolation caused by the pandemic has directly affected people’s mental health. However, adolescents showed different results. Before the outbreak of COVID-19, the experience rate of depressive mood in Korean adolescents was 22.2% for men and 34.6% for women, but after the outbreak, it was 20.1% for men and 30.7% for women in 2020 and 22.4% for men and 31.4% for women in 2021—a decrease compared to 2019, before the outbreak of COVID-19 [2]. From January 2020 to December 2020, when COVID-19 started spreading, the suicide rate of Koreans decreased by 1.2 per 100,000 people compared to 2019, while the suicide rate of teenagers increased by 0.6 per 100,000 people. Accordingly, the proportion of suicide deaths in the total number of deaths increased from 37.5% to 41.1% [3]. Previous studies have shown that an increase in depression, anxiety, and substance use disorders can ultimately develop into suicidal behavior [4] and that there is a delay between the peak of the pandemic and the onset of suicidal behavior [5]. This behavior may be avoided if people at risk of suicide during that delay are spotted and helped by appropriate intervention. From this theoretical background, it is notable that the suicide rate increased despite a decrease in the experience rate of depressive mood in adolescents. No clear evidence suggests that this increase in youth suicide is caused by COVID-19, but there is no ground to exclude the impact of mental health caused by it, given the high relationship between suicide and mental health disorders such as depression. Furthermore, if this increase in suicide rate is significantly related to COVID-19 it can mean that mental health issues, such as increased depressive feelings during the COVID-19, develop rapidly into suicidal behavior in adolescents; thus, immediate intervention is necessary. Therefore, this study was conducted to understand the changes in depression among adolescents after COVID-19 and to determine the effect of increased depressive feelings after COVID-19 compared to the previously known relationship between depression and suicidal behavior. It can be used as basic data to establish youth mental health-related measures and prevent suicide in the ongoing COVID-19 era. The hypothesis of this study is as follows.

**Hypothesis 1 (H1).** 
*Even in adolescents without a depressive mood, a temporary increase in depressive feelings caused by COVID-19 will affect suicidal behavior.*


**Hypothesis 2 (H2).** 
*Depressive mood and increase in Depressive feelings during the COVID-19 have a synergistic effect.*


## 2. Materials and Methods

### 2.1. Data Source

This study was conducted using secondary data downloaded from the 17th (2021) Korea Youth’s Risk Behavior Web-Based Survey (KYRBS) website. KYRBS is a survey carried out by Korea Centers for Disease Control and Prevention (Approval No. 117058) with the aim of understanding the current status and trend of Korean youth health behavior. The total population of the 17th KYRBS was 2,629,588 students from 5642 schools nationwide as of April 2021, and samples were extracted by stratified colony sampling method (800 schools, 59,426 students). All survey items, including height and weight, were self-reported and were collected through an online questionnaire using a computer or a mobile device (tablet PC, smartphone). All study subjects agreed to participate in the survey using this online questionnaire. For more details on the questionnaire, we attach the CHERRIES guidelines. (Appendix A).

### 2.2. Research Subjects

Among the 54,848 respondents of the 17th KYRBS, we decided to conduct this study on high school students (24,833). Among the 24,833 high school respondents, we excluded 4 subjects who did not respond to the question of whether their depressive feeling changed during the COVID-19. Our final research subjects were 24,829 students, which we classified into male (12,813) and female (12,016) for analysis (Figure 1).

### 2.3. Independent Variables

The 17th (2021) Korea Youth’s Risk Behavior Web-Based Survey (KYRBS) defines depressive mood as feeling sad or hopeless to the extent of interfering with daily activities for 2 weeks within the past 12 months and asks the respondents to answer “yes” or “no”. Increase in depressive feeling is evaluated as “How much has your depressive feeling changed compared to pre-COVID19 pandemic?”. We classified ‘greatly increased’ and ‘increased’ as ‘increase in depressive feeling’ and ‘unchanged’, ‘decreased’, ‘greatly decreased’ as ‘no increase in depressive feeling’. To evaluate the effects of depressive mood and increase in depressive feeling on suicidal behavior, we classified the responses in four different groups by relevant variables: ‘no depressive mood and no increase in depressive feeling (group A)’, ‘no depressive mood and increase in depressive feeling (group B)’, ‘depressive mood and no increase in depressive feeling (group C)’, ‘depressive mood and increase in depressive feeling (group D)’.

### 2.4. Dependent Variables

The Dependent variable of this study is ‘presence of suicidal behavior’. If the respondent has experienced any of suicidal ideation, planning, or attempt, they were classified as ‘present suicidal behavior’, and ‘no suicidal behavior’ when they had none of the above experiences. 17th (2021) Korea Youth’s Risk Behavior Web-Based Survey (KYRBS) asks about suicidal ideation as ‘Have you ever seriously considered suicide within the last 12 months?’, suicidal planning as ‘Have you ever made specific plans to commit suicide within the last 12 months?’, suicidal attempt as ‘Have you ever tried to commit suicide within the last 12 months?’, and respondents answered, ‘yes’ or ‘no’.

### 2.5. Confounding Variables

The confounding variables of depressive mood and suicide behavior were classified into demographic characteristics {grade (10th, 11th, 12th), academic performance (high, middle, low), economic status (high, middle, low), economic deterioration after COVID-19 (yes, no)}, and health behavioral characteristics {change in physical activity after COVID-19 (increase, same, decrease), current smoking (no, yes), current drinking (no, yes), perceptual body image (skinny, normal, fat), and subjective sleep satisfaction (yes, no)}, which were controlled for in the statistical analysis.

#### 2.5.1. Demographic Characteristics

Academic performance and economic status were classified into three groups: high, middle, and low. If the respondents’ answers to the question were ‘high’ or ‘medium high’, they were classified into the ‘high group’, if their answer was ‘medium’, they were classified into the ‘middle group’, and if their answers were ‘medium low’ or ‘low’, they were classified into the ‘low’ group. For the case of economic deterioration after COVID-19, if the respondents’ answers were ‘strongly agree’ or ‘agree’, they were classified into ‘economic deterioration after COVID-19’ while ‘disagree’ and ‘strongly disagree’ were classified into ‘no economic deterioration after COVID-19’. Additionally, in order to analyze the relationship between changes in economic status and suicide behavior after COVID-19, as well as economic status, we divided the respondents into six groups. The type of residence was classified into five types: living with family, living with relatives, boarding/living alone, dormitory, and childcare facilities. We categorized residency into 2 groups: living with family and not living with family.

#### 2.5.2. Health Behavioral Characteristics

For changes in physical activity after COVID-19, we categorized the respondents into 3 groups: ‘highly increased’ and ‘increased’ into ‘increase’ group, ‘same’ into ‘same’ group, ‘decreased’ and ‘highly decreased’ into ‘decrease’ group. Currently smoking was defined as having smoked one of cigarettes, liquid electronic cigarettes, or electronic cigarettes within the last month. Currently non-smoking was defined as having never smoked before or smoked none of the above within the last month. Currently drinking was defined as having drunk alcohol within a month, while ‘not drinking’ was defined as having never drunk before or not having drank within the last month. For the question “What do you think about your body shape?”, if the respondents’ answers were ‘slightly skinny’ or ‘skinny’, we classified them into ‘skinny’ group, and if the respondents’ answers were ‘normal’, we classified them into ‘normal’ group. If the answers were ‘slightly fat’, or ‘fat’, they were categorized into the ‘fat’ group. In case of subjective sleep satisfaction, ‘satisfied’ and ‘very satisfied’ were categorized into the ‘satisfied’ group, and ‘so-so’, ‘not satisfied’, and ‘not at all satisfied’ were grouped into ‘not satisfied’. We have attached the details of the questionnaire of health behavioral characteristics to the Appendix A.

### 2.6. Statistical Analysis

Korea Youth’s Risk Behavior Web-Based Survey (KYRBS) data used complex sampling design [6]. Thus, we conducted complex sampling analysis by applying strata, cluster, and weight to the data. All variables used in this study are categorical variables. First, using the complex samples frequencies procedure to confirm the general properties of the respondents. After that, we defined sex as stratified variable in complex sample crosstabs procedure to analyze the difference in suicidal behavior in each variable by sex, using chi-squared test. A complex sample logistic regression analysis was conducted to analyze the effect of depressive mood and increased depressive feeling on the risk of suicidal behavior after COVID-19. Variables with a significant difference (*p* < 0.05) in the chi-square test were controlled as confounding variables. The main results are expressed as odds ratios (ORs), significance levels (*p*-value), and confidence interval (CI), and 95% confidence interval (*p* < 0.05) was interpreted as statistically significant. All data analyses were carried out with IBM SPSS Statistics version 26.0 (IBM Corporation, Armonk, NY, USA).

## 3. Results

When suicidal behavior was analyzed according to gender, the frequency of suicidal behavior was higher in women (N = 1929, 15.9%) than in men (N = 1196, 9.6%). Table 1 shows the analysis of the presence of a suicidal behavior of both male and female subjects according to the general characteristics. Comparing three year groups, there was no significant difference in both male and female subjects based on a suicidal behavior. However, both male and female had a significant difference in a suicidal behavior according to the academic performance. (male *p* = 0.016, female *p* < 0.001) There were significant difference in a suicidal behavior depending on the type of residence, (male *p* = 0.008, female *p* < 0.001) there also was a significant difference in suicidal behavior based on other variables such as current economic status, post-COVID-19 economic deterioration, post-COVID-19 physical activity, current tobacco use, current drinking habit, perceptual body image, sleep satisfaction, increased depressive feeling and depressive mood (both male and female variable *p* < 0.001).

Table 2 summarizes the relationship between suicidal behavior and depressive mood and increased depressive feelings from the results of logistic regression analysis using three models. Model 1 did not include confounding variables; the group with neither a depressive mood nor an increase in depressive feeling was taken as the reference. Compared to the reference group, the odds of male in group B, C, D were 3.96, 11.36 and 22.30, and the odds of female in group B, C, D were 3.23, 10.22 and 19.53, respectively. Model 2 is a model where variables that induced significant difference in chi-square test such as academic performance, economic condition, post COVID-19 economic deterioration, post COVID-19 physical activity, current tobacco use, current drinking habit, perceptual body image, sleep satisfaction were controlled. After the control, compared to the reference group, the odds of male in group B, C, D were 3.57, 10.20 and 18.30, and the odds of female in group B, C, D were 2.88, 8.15 and 14.87, respectively.

## 4. Discussion

A suicidal behavior of both male and female subjects shows that females present more suicidal behavior than males. It is well known that suicide thoughts and attempts are higher in females, and our study also showed the same results [7]. A study related to suicide attempters and suicides in children and adolescents found that 80–90% had a diagnosable mental illness at the time of the suicide attempt and showed a particularly high coexistence rate with depression [8]. In other words, mental illness is closely related to suicide and mood disorders—especially depression—[9]. After the peak of the pandemic, the number of people complaining of depressive mood along with an increase in depressive feelings grew, in spite of a considerable amount of time is required for mental health conditions such as depression to transition into suicidal behavior [5]. However, in 2020, when COVID-19 lasted for approximately a year, it was announced that the suicide rate of the entire Korean population decreased, while the number of youth suicides increased. In adolescents, studies have shown that suicide attempts are impulsive rather than planned [10]. It has also been shown that adolescents who attempt suicide are more impulsive than those who do not [11], which means that adolescents who are more emotional than adults are more likely to attempt or commit suicide impulsively when their depressive feeling increases, and the time interval between depressive mood and depression and suicide may be short. Therefore, this study attempted to analyze the role of the change in depressive feelings caused by COVID-19 in the relationship between depressive mood and suicidal behavior, using high school students who have had no depressive mood and no increase in depressive feelings after COVID-19 as a reference group. Similar to previous studies, the results of the study showed that depressive feelings increased the risk of suicidal behavior, and in group C with only depressive mood, the odds ratio of model 1 was 11.36 for men and 10.22 for women, showing a 10-fold increase in the risk of suicidal behavior compared to group A Moreover, in the case of group B, which had no depressive mood but increased depressive feelings during the COVID-19, the odds ratio was 3.96 for men and 3.23 for women. In other words, due to the relative increase in depressive feelings caused by COVID-19, suicidal behavior significantly increased without depressive mood. Group D showed the highest odds ratio related to suicidal behavior, at 22.30 for men and 19.53 for women. When depressive feelings increase in the presence of depressive mood, the odds ratio also increases significantly, which shows that the increase in depressive feelings during the COVID-19 has a synergistic effect on suicidal behavior. We calculated the relative excess risk due to interaction (RERI) using the odds ratio from model 2. The calculated values of RERI were both greater than 0–5.53 for men and 4.83 for women—which suggests that depressive mood and an increase in depressive feelings during the COVID-19 have a synergic effect rather than an additive effect [12]. Therefore, considering that suicidal behavior increased without a long-time interval when depressive feelings increased during the COVID-19, the mental health management of adolescents’ increased depressive feelings in a social crisis is highly important and should be reflected in related policies.

In this study, when analyzing the relationship between suicidal behavior and variables that are known to influence suicidal behavior—such as grade, academic performance, economic status, the presence of economic deterioration after COVID-19, smoking, drinking, perceptual body image, and sleep satisfaction—all variables except grade were related to suicidal behavior. Previous studies have shown that individual factors, including gender, self-esteem, atrophy, aggression, experiences of sexual harassment, and experiences of delinquency; family factors, including relationships with parents, physical abuse, emotional abuse, neglect, household economic status, and solutions of family conflict; and school factors, including academic stress, experiences of being a victim of school bullying, and attachment toward friends, are related to suicidal behavior [13]. However, this study utilized secondary data that had no relevant variables among the ones listed above; thus, we could not analyze and adjust for the relationship between these variables and suicidal behavior. Follow-up studies including these variables should be conducted to understand the relationship with suicidal behavior, and effective suicide prevention and intervention should be conducted with specific plans for overall lifestyle as well as family support when establishing youth suicide prevention policies.

Among these variables, it was confirmed that the subjective economic status of students and the presence of economic deterioration after COVID-19 were significantly related to suicidal behavior. Compared to other groups, when economic status was “low”, both male and female students had the highest suicide rate. In addition, in all groups, when economic status deteriorated, the proportion of students who experienced suicidal behavior was higher than when economic status did not deteriorate. Previous studies have shown that low economic conditions are highly related to depressive mood and suicidal thoughts and that economic deterioration also increases depressive mood and suicidal thoughts in adolescents [14]. This study shows results similar to those of previous studies, but it focuses on whether economic status worsened after COVID-19. If social crises such as COVID-19 recur, institutional measures should be prepared to help adolescents suffering from economic difficulties and deterioration.

Compared to male students, the proportion of female students who thought that they had gained weight (or were obese) was higher, and suicidal behavior for both male and female students increased significantly when they perceived themselves as having gained weight. Previous studies conducted in the United States have shown that self-awareness of body type became clear as weight increased and that obesity exceeding the standard of overweight was related to suicidal behavior [15]. Furthermore, differences can be observed according to race, which is consistent with this study as it shows that self-body type recognition has a greater effect on suicidal behavior than actual BMI [16]. Therefore, it seems that educating students to have the right perception of their body type can also help prevent suicidal behavior.

This study has several limitations because it used secondary data and was designed as a cross-sectional study. First, as this was a cross-sectional study, it was difficult to grasp the cause-and-effect relationship between confounding variables such as depressive mood and suicidal behavior. Furthermore, as mentioned in the discussion, mental illness is closely related to suicide. However, risk factors for suicide behavior such as mental and physical health history were not included in the questionnaire, so we could not deal with them. Second, depressive mood and change of depressive feeling, a major independent variable, was measured using a subjective questionnaire rather than a structured one such as the Patient Health Questionnaire-9 (PHQ-9). Third, in addition to the confounding variables used in this study, important variables such as relationships with classmates, family environments, and parents were omitted. Therefore, confounding variables were not sufficiently accounted for, and the correlation between each variable and suicidal behavior was not identified. Fourth, in the research method, suicide thoughts, plans, and attempts were grouped into one variable, namely, suicidal behavior, and then analyzed. This process has its limitations as it simplifies the complex relationship leading to suicidal thoughts and planning attempts.

Despite these limitations, this study had the advantage of a systematic sampling process, and it used a high representation of samples as nationwide survey data. Moreover, regarding the relationship between existing depressive mood and suicidal behavior, this study is significant in showing a synergistic relationship between the relatively short-term effect of increasing depressive feelings during the COVID-19 and depressive mood. These results suggest that early intervention for depressive mood is important in establishing measures to prevent youth suicide as this can happen in a relatively short period of time.

## 5. Conclusions

The results of this study showed that an increase in depressive feelings during the pandemic is significantly related to suicidal behavior. In particular, suicidal behavior also increased significantly even when subjects initially without depressive mood experienced an increase in depressive feelings after COVID-19. This suggests that the short-term increase in depressive feelings caused by pandemics increases the risk of suicidal behavior in adolescents, whose suicide tends to be more impulsive than planned. In addition, this study showed that an increase in depressive feelings in subjects with depressive mood results in a greater risk of suicidal behavior due to its synergistic effect. This finding emphasizes the importance of mediating depressive mood as early as possible to prevent youth suicide. It is necessary for follow-up studies in the future to identify the risk factors of adolescents’ depression so that the risk group of youth suicide can be detected early in social crises, such as pandemics, and appropriate intervention can be made.

## Figures and Tables

**Figure 1 ijerph-19-14868-f001:**
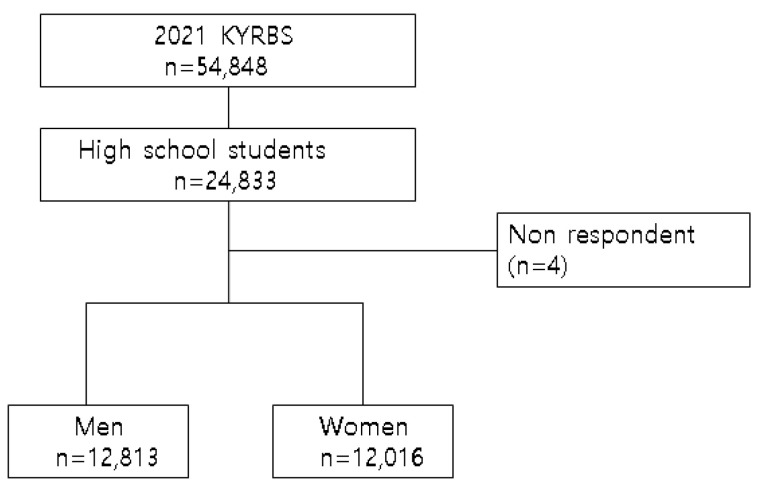
Subject selection.

**Table 1 ijerph-19-14868-t001:** Sociodemographic characteristics of participants and suicidal behavior.

Variables	Suicidal Behavior							
	Male (n = 12,813)				Female (n = 12,016)			
	No	Yes	X^2^	*p*-Value	No	Yes	X^2^	*p*-Value
Grade, n (%)			2.88	0.331			0.14	0.946
10th	4064 (91.00 *,32.00 **)	392 (9.00 *,29.60 **)			3361 (83.90 *,31.90 **)	642 (16.10 *,32.30 **)		
11th	4044 (90.10 *,33.70 **)	433 (9.90 *,34.70 **)			3477 (84.20 *,33.90 **)	692 (15.80 *,33.60 **)		
12th	3509 (90.00 *,34.30 **)	371 (10.00 *,35.70 **)			3249 (84.20 *,34.30 **)	595 (15.80 *,34.10 **)		
Academic performance, n (%)			11.31	0.016			47.25	<0.001
High	3666 (91.10 *,31.40 **)	341 (8.90 *,29.00 **)			2999 (86.30 *,29.80 **)	485 (13.70 *,25.10 **)		
Middle	3662 (91.10 *,31.50 **)	345 (8.90 *,29.00 **)			3499 (85.50 *,35.00 **)	591 (14.50 *,31.40 **)		
Low	4289 (89.20 *,37.10 **)	510 (10.80 *,42.00 **)			3589 (81.10 *,35.20 **)	853 (18.90 *,43.40 **)		
Economic status and deterioration post-COVID 19, n (%)			91.67	<0.001			194.68	<0.001
High, no deterioration	3398 (91.10 *,29.70 **)	323 (8.90 *,27.40 **)			2742 (87.00 *,28.50 **)	423 (13.00 *,22.40 **)		
High, deterioration	868 (88.60 *,7.50 **)	103 (11.40 *,9.10 **)			498 (80.00 *,4.90 **)	122 (20.00 *,6.50 **)		
Middle, no deterioration	3891 (92.40 *,33.60 **)	315 (7.60 *,25.90 **)			3803 (85.30 *,37.50 **)	554 (12.70 *,28.80 **)		
Middle, deterioration	1882 (90.50 *,16.00 **)	185 (9.50 *,15.70 **)			1807 (82.30 *,17.40 **)	387 (17.70 *,19.90 **)		
Low, no deterioration	644 (88.60 *,5.40 **)	87 (11.40 *,6.50 **)			461 (76.40 *,4.40 **)	144 (23.60 *,7.20 **)		
Low, deterioration	934 (82.70 *,7.80 **)	183 (17.30 *,15.40 **)			776 (71.60 *,7.20 **)	299 (28.40 *,15.20 **)		
Type of residence, n (%)			17.68	0.003			39.24	<0.001
With family	10,682 (90.70 *,92.70 **)	1060 (9.30 *,89.20 **)			9470 (84.60 *,95.30 **)	1746 (15.40 *,91.50 **)		
else	935 (86.40 *,7.30 **)	136 (13.60 *,10.80 **)			617 (74.70 *,4.70 **)	183 (25.30 *,8.50 **)		
Post-COVID 19 physical activities, n (%)			22.95	<0.001			31.44	<0.001
Increase	2557 (89.80 *,21.70 **)	279 (10.20 *,23.00 **)			987 (80.60 *,9.50 **)	241 (19.40 *,12.10 **)		
Same	4102 (92.10 *,35.10 **)	340 (7.90 *,28.50 **)			3406 (86.60 *,32.80 **)	537 (13.40 *,26.90 **)		
Decrease	4958 (89.30 *,43.20 **)	577 (10.70 *,48.50 **)			5694 (83.40 *,57.70 **)	1151 (16.60 *,61.00 **)		
Tobacco use, n (%)			80.79	<0.001			199.35	<0.001
No	10,387 (91.30 *,89.40 **)	960 (8.70 *,80.20 **)			9720 (85.30 *,96.60 **)	1692 (14.70 *,88.10 **)		
Yes	1230 (83.40 *,10.60 **)	236 (16.60 *,19.80 **)			367 (60.20 *,3.40 **)	237 (39.80 *,11.90 **)		
Alcohol use, n (%)			68.30	<0.001			160.21	<0.001
No	9545 (91.50 *,82.10 **)	860 (8.50 *,72.00 **)			8915 (85.80 *,88.80 **)	1485 (14.20 *,77.50 **)		
Yes	2072 (85.70 *,17.90 **)	336 (14.30 *,28.00 **)			1172 (72.40 *,11.20 **)	444 (27.60 *,22.50 **)		
Perceptual body image, n (%)			247.75	<0.001			501.37	<0.001
Skinny	8257 (92.90 *,70.90 **)	603 (7.10 *,50.90 **)			6004 (89.70 *,59.50 **)	688 (10.30 *,36.10 **)		
Normal	2457 (87.60 *,21.20 **)	349 (12.40 *,28.20 **)			3086 (81.60 *,30.60 **)	712 (18.40 *,36.50 **)		
Fat	903 (78.10 *,7.90 **)	244 (21.90 *,20.90 **)			997 (65.80 *,10.00 **)	529 (34.20 *,27.40 **)		
Sleep satisfaction, n (%)			31.87	<0.001			44.04	<0.001
Satisfied	2543 (93.10 *,22.20 **)	180 (6.90 *,15.50 **)			1580 (89.20 *,15.60 **)	196 (10.80 *,10.00 **)		
Not satisfied	9074 (89.60 *,77.80 **)	1016 (10.40 *,84.50 **)			8507 (83.20 *,84.40 **)	1733 (16.80 *,90.00 **)		
Increased depressive feeling and depressive mood, n (%)			1592.02	<0.001			1918.29	<0.001
Group A	7509 (97.60 *,64.50 **)	182 (2.40 *,14.70 **)			4828 (96.60 *,47.20 **)	179 (3.40 *,8.80 **)		
Group B	1975 (91.20 *,17.40 **)	188 (8.80 *,15.70 **)			2778 (89.70 *,28.10 **)	318 (10.30 *,17.00 **)		
Group C	1075 (78.40 *,9.10 **)	277 (21.60 *,23.40 **)			935 (73.50 *,9.10 **)	335 (26.50 *,17.40 **)		
Group D	1058 (64.90 *,9.10 **)	549 (45.10 *,46.30 **)			1546 (59.20 *,15.60 **)	1097 (40.80 *,56.80 **)		

* row%, ** column.

**Table 2 ijerph-19-14868-t002:** Relationship between the Increased Depressive Feeling and Depressive Mood and Suicidal Behavior.

Variables	Male		Female	
	Model 1	Model 2	Model 1	Model 2
	Odds Ratio ^†^ (95% CI)		Odds Ratio ^†^ (95% CI)	
Group B	3.96 * (3.09–5.08)	3.57 * (2.74–4.64)	3.23 * (2.62–3.99)	2.88 * (2.32–3.58)
Group C	11.36 * (9.15–14.10)	10.20 * (8.18–12.72)	10.22 * (8.38–12.48)	8.15 * (6.58–10.09)
Group D	22.30 * (18.08–27.49)	18.30 * (14.70–22.78)	19.53 * (16.38–23.38)	14.87 * (12.36–17.88)

Model 1: crude odds ratio, Model 2: model 1 + controlled variables (academic performance, economic status and deterioration post-COVID 19, type of residence, post-COVID 19 physical activity, tobacco use, alcohol use, perceptual body image, sleep satisfaction) * *p* < 0.001; ^†^ group A is reference group.

## Data Availability

Publicly available datasets were analyzed in this study. This data can be found here: [https://www.kdca.go.kr/yhs/] accessed on 1 September 2022.

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
