# Peer review of "Effect of Increased Depressive Feelings during the COVID-19 Pandemic on the Association between Depressive Mood and Suicidal Behavior: Using the 17th (2021) Korea Youth Risk Behavior Web-Based Survey (KYRBS)"

_ijerph, 2022, doi:10.3390/ijerph192214868_

Round 1

Reviewer 1 Report

The study entitled “Moderating Effect of Depressive Feeling due to COVID-19 on 2 the Association between Depressive Mood and Suicidal 3 Behavior: Using the 17th (2021) Korea Youth Risk Behavior 4 Web-Based Survey (KYRBS)” addresses an important topic with regard to factors that likely associate with suicidal behaviour in adolescents in Korea; in particular the study aimed to examine the impact of increased depressive feelings due to the COVID-19 pandemic on the suicidal behavior of Korean high school students using the 17th Korea Youth Risk Behavior Web-Based Survey (KYRBS).

Overall, the study is interesting, however some specific areas to attend are noted below.

1.      The authors have taken into consideration several socio-demographic factors, however it would be important if they could include data regarding adolescents’ mental and physical health history, such as possible diagnosis (psychiatric / or comorbidities), previous suicidal behaviour, previous risk behaviors etc. If this was not possible due to the nature of the study (i.e, secondary data) it should be mentioned in the Limitations.

2.      The change in the depressive feelings compared to pre-COVID19 pandemic has been defined according to one self-report item that the students answered in April 2021. This alone may be a source of significant bias in the data and hence in the conclusions of the study.

3.      It is suggested that more information regarding logistic analysis is provided. As the count of the cases is rather low especially in the reference group A, have the authors used any exact methods to address possible issues?

4.      Please change p=0.000 to p<0.001 in Table 1

Reviewer 2 Report

The manuscript is interesting but needs some improvements

Major comments

The title should be changed, the term “moderating effect” might be misleading, and also “depressive feelings due to COVID-19” does not seem to reflect item analyzed. I suggest to start with “Effect of increased depressive feelings during the pandemic on the association between …”

Introduction: is it real (line 38) that in 2021 depressive mood was 31.4% in men and 22.4% in women (a radical change with respect to previous years), or are data inverted?

Results: all analyses are stratified by gender. That’s fine, but at the beginning of the Results section the overall prevalence of suicidal behavior separately in males in females should be reported (and briefly commented in the Discussion)

Discussion: there is some error at the end of page 6 that might be highly confusing for the reader. In fact, in model 1 the OR associated with depressive mood reported in the text seem to be inverted between males and females (compared to numbers reported in the Table). Furthermore, in the following lines the comment “In other words, due to the relative increase in depressive feelings caused by COVID-19, suicidal behavior significantly increased without depressive mood” should be deleted, or otherwise preceded by reporting the OR for group B (3.96 in males and 3.23 in females).

Lastly, I’m not a native English speaker, but the manuscript should be extensively copyedited. Only as examples:

First line of the Data source section of Methods, please delete “which the source was”

Last line of the Confounding variables of Methods, please use “which were accounted for in the statistical analysis” instead of “which were corrected for analysis”. Similar corrections should be applied thorough the paper.

Results, page 5, description of Table 2, please report “Model 1 did not include confounding variables; the group with neither a depressive mood nor an increase in depressive feeling was taken as the reference.” instead of “Model 1 is a model where the confounding variable was not revised, and was based on the group that responded that there was neither a depressive mood nor an increase in depressive feeling.”

Reviewer 3 Report

The current manuscript described a study which assess the association between self-reported depressive symptoms and suicide ideation/planning/attempt, using data from the 17th (2021) Korea Youth's Risk Behavior Web-Based Survey.

My overall appraisal is that the manuscript lacks several pieces of key information about the sampling design and statistical analysis. This is a substantial issue, because are the potential strengths of general population surveys. In this regard, is important to note that the survey was carried on using online tools, however no further details are provided to acknowledge the extent of the application of the CHERRIES guidelines.

In addition to these limitations, I believe that the title and the corresponding aim of the study doesn’t correspond to the methods that were used. It seems that authors only performed a multivariate logistic regression to assess the association between a four-group classification based on two questions about depression. However, further procedures are needed to test a moderation hypothesis. Also, the title is a little confusing because it states that depression moderates the association between depression and suicide behavior.

There are some other additional issues that can be improved:

The first it is that there is no substantial background to propose a moderation model. It is important to substantially support this hypothesis using previous findings and theoretical contributions.

Information about sampling methods and recruitment should be further explained. More detail is required.

In line 71 Please change self-written to self-reported.

Taking into consideration that this was a web survey authors should consider including at least some of the CHERRIES statement items.

Figure 1. This diagram is a little misleading. I don´t quite get why the 2021 Korean Youth was added (the one that is about 2.6 million). A common practice when we display this kind of data is to start with the number of recruited or enrolled participants.

Four group depression classification requires theoretical or empirical foundation. As it is, it may not be compatible with current methods for depression screening or assessment.

Please state in the analysis section how the confounding variables were corrected or controlled. Also, I believe that controlled for is a more proper term than corrected for.

Where does the health behavior characteristics question comes from? Please add any substantial reference.

It would be useful to add a reference to additional information regarding sampling design.

Please describe which methods was performed to estimate and correct for standard errors

In every table, please display chi squared test result. Also change the percentages to column, currently row percentages are display and that is a little confusing.

In page 5 line 167 please change "was not revised" to "was not included"

In Table 2 reporting Group A is not necessary, please only indicate that it is the comparison group. There were any pairwise difference? This could be very informative as it seems that these groups could be considered as levels of severity

I made no further review to the study discussion as it seems that the main objective was confusing and the statistical approach was unable to respond a moderation hypothesis, therefore results and discussion can’t provide trustworthy information.

Round 2

Reviewer 2 Report

The manuscript now needs only a minor copyediting, e.g.:

page 4, line 133 "Variables with a significant..." instead of "For variables with a significant..."

Discussion 1st paragraph: please rearrange the sentence about mental illness and suicide

Conclusion, first sentence: better "during the pandemic" instead of "due to COVID-19"

Lastly, in the first round of review I suggested to include the prevalence of suicidal behaviour by sex. I would keep it in the text, without adding Figure 2

Reviewer 3 Report

Authors have conducted several changes in the manuscript. However, I there are two critical concers, that from my point of view, limit the contribution of this work to the current literature about suicide.

My first concern relates to the aim of the study, it seems that it was intended to associate depression and suicide. This association is already established in the literature. 

 My second concern relates to the classification of individuals in four groups. This classification lacks of empirical or theorical foundation, therefore the interpretation of the results may be limited.

An additional concern is the use of the terms "depressive mood" and "depressive feelings" in an ambigous way. I find it hard to interpret if these terms are synonims or refer to diffent levels of depression symptoms.
